METHODS AND RESOURCES

# Highly efficient CRISPR-mediated gene editing in a rotifer

**Haiyang Feng, Gemma Bavister, Kristin E. Gribble**  *, **David B. Mark Welch** *

Josephine Bay Paul Center for Comparative Molecular Biology and Evolution, Marine Biological Laboratory, Woods Hole, Massachusetts, United States of America

* kgribble@mbl.edu (KEG); dmarkwelch@mbl.edu (DMW)

## Abstract

Rotifers have been studied in the laboratory and field for over 100 years in investigations of microevolution, ecological dynamics, and ecotoxicology. In recent years, rotifers have emerged as a model system for modern studies of the molecular mechanisms of genome evolution, development, DNA repair, aging, life history strategy, and desiccation tolerance. However, a lack of gene editing tools and transgenic strains has limited the ability to link genotype to phenotype and dissect molecular mechanisms. To facilitate genetic manipulation and the creation of reporter lines in rotifers, we developed a protocol for highly efficient, transgenerational, CRISPR-mediated gene editing in the monogonont rotifer *Brachionus manjavacas* by microinjection of Cas9 protein and synthetic single-guide RNA into the vitellaria of young amictic (asexual) females. To demonstrate the efficacy of the method, we created knockout mutants of the developmental gene *vasa* and the DNA mismatch repair gene *mlh3*. More than half of mothers survived injection and produced offspring. Genotyping these offspring and successive generations revealed that most carried at least 1 CRISPR-induced mutation, with many apparently mutated at both alleles. In addition, we achieved precise CRISPR-mediated knock-in of a stop codon cassette in the *mlh3* locus, with half of injected mothers producing F2 offspring with an insertion of the cassette. Thus, this protocol produces knockout and knock-in CRISPR/Cas9 editing with high efficiency, to further advance rotifers as a model system for biological discovery.

## Introduction

Rotifers are microscopic invertebrates found globally in aquatic habitats; at times they are the most abundant animals in some freshwater ecosystems. Rotifers comprise a phylum within the protostome clade Gnathifera [1,2] and are by far the most experimentally tractable of this early branching example of metazoan evolution [3]. Monogonont rotifers, one of the two major groups of Rotifera, have long been used in studies of evolution, limnology, ecology, and toxicology [4,5]. One of the most well-studied genera, *Brachionus*, is cultured at large-scale worldwide as live feed for hatchery rearing in aquaculture, leading to intense interest in improving their nutritional value and energy content. In recent years, *Brachionus* has been an important representative in studies of metazoan body plan evolution [1,5] and has reemerged as an

**Data Availability Statement:** All relevant data are within the paper and its Supporting Information files.

**Funding:** This project was supported by a grant from the Marine Biological Laboratory to KEG and

DMW. KEG was supported by Grant R21AG067034 from the National Institute on Aging. The funders had no role in study design, data collection and analysis, decision to publish, or preparation of the manuscript.

**Competing interests:** The authors have declared that no competing interests exist.

**Abbreviations:** CPP, cell-penetrating peptide; HDR, homology-directed repair; HRM, high-resolution melt; IO, Instant Ocean; NHEJ, nonhomologous end joining; PAM, protospacer adjacent motif; RNAi, RNA interference; sgRNA, single-guide RNA; ssDNA, single-stranded DNA; VNTR, variable number tandem repeat.

alternative model in translational research areas such as aging and maternal effects [6,7]. However, a lack of gene editing tools, transgenic lines, and molecular reporter strains has hindered research on molecular mechanisms using *Brachionus* or other rotifers.

*Brachionus* is an attractive experimental system not only because of the animals' small size, transparency, ease of culturing, and short generation time but also because of their particular life cycle: Reproduction is usually amictic (asexual), with diploid females producing daughters by mitotically derived embryos, resulting in clonal cultures. In response to species-specific environmental cues (often crowding), these amictic females produce mictic daughters that undergo meiosis in their ovaries. Unfertilized eggs develop into haploid males, which mate with mictic females to fertilize undeveloped haploid eggs, producing overwintering resting eggs that hatch as amictic females [8,9]. This has led to using rotifers for extensive research on the environmental cues and molecular controls on inducible sexual reproduction and for investigation on the evolutionary fitness effects and ecological consequences of this bet hedging life history strategy [10–12]. Alternating asexual and sexual generations permits laboratory experimentation with asexual lineages in which all individuals are genetically identical, or with sexual reproducing, genetically diverse populations. The absence of genetic manipulation methods has slowed mechanistic research on these topics, however.

Existing genomic resources for *Brachionus* species include several sequenced genomes and transcriptomes [13–17]. A transfection-based RNA interference (RNAi) protocol is available, though it has not been shown to act transgenerationally [18,19]. Cell-penetrating peptides (CPPs) [20] and lipofection reagents [21] facilitate uptake of plasmid DNA by *Brachionus*, but to date, these techniques have not been shown to penetrate germ tissue. In 2019, CRISPR/Cas9 activity was demonstrated in rotifers by electroporating Cas9 and single-guide RNAs (sgRNAs) targeted to a cytochrome P450 gene into *Brachionus koreanus*, deep sequencing PCR products generated from DNA of electroporated animals, and detecting indels in a small fraction of gene copies consistent with CRISPR activity [22]. However, as with RNAi and CPP transfection, this method did not affect the germ line and thus did not result in a stable mutated lineage.

Here, we develop and describe a CRISPR/Cas9 protocol that delivers Cas9 and sgRNA through microinjection, achieving transgenerational knockout and knock-in mutations in offspring at high efficiency and enabling the establishment of stable, clonal, mutant lines.

## Results

### General approach

We immobilized neonates (newly hatched amictic females) by grasping the apical corona with a holding needle and injected Cas9 protein and sgRNA into the vitellarium, a reproductive tissue that provides material to developing oocytes, analogous to the nurse cells of *Drosophila* or rachis of *C. elegans* (Fig 1). Using this approach, 50% to 66% of the injected neonates survived to produce offspring. We call injected individuals "mothers," in which Cas9/sgRNA could have been delivered to oocytes. After injection, each mother was transferred to an individual well of a tissue culture plate to reproduce. The offspring of injected mothers were designated as the F0 generation, consisting of asexually produced individuals potentially altered by CRISPR at the single-cell stage and/or at later developmental stages in which Cas9 and sgRNA may have directly affected the F0 germline. We transferred the first 4 to 6 F0 individuals produced by each mother to discrete wells of a tissue culture plate. Each F0 female asexually produced multiple F1 offspring, which we subsequently transferred individually or in pools to fresh wells. To establish whether editing had occurred and to define the range of mutation types, we genotyped F0, F1, and subsequent asexual generations (Fn) by PCR amplification of

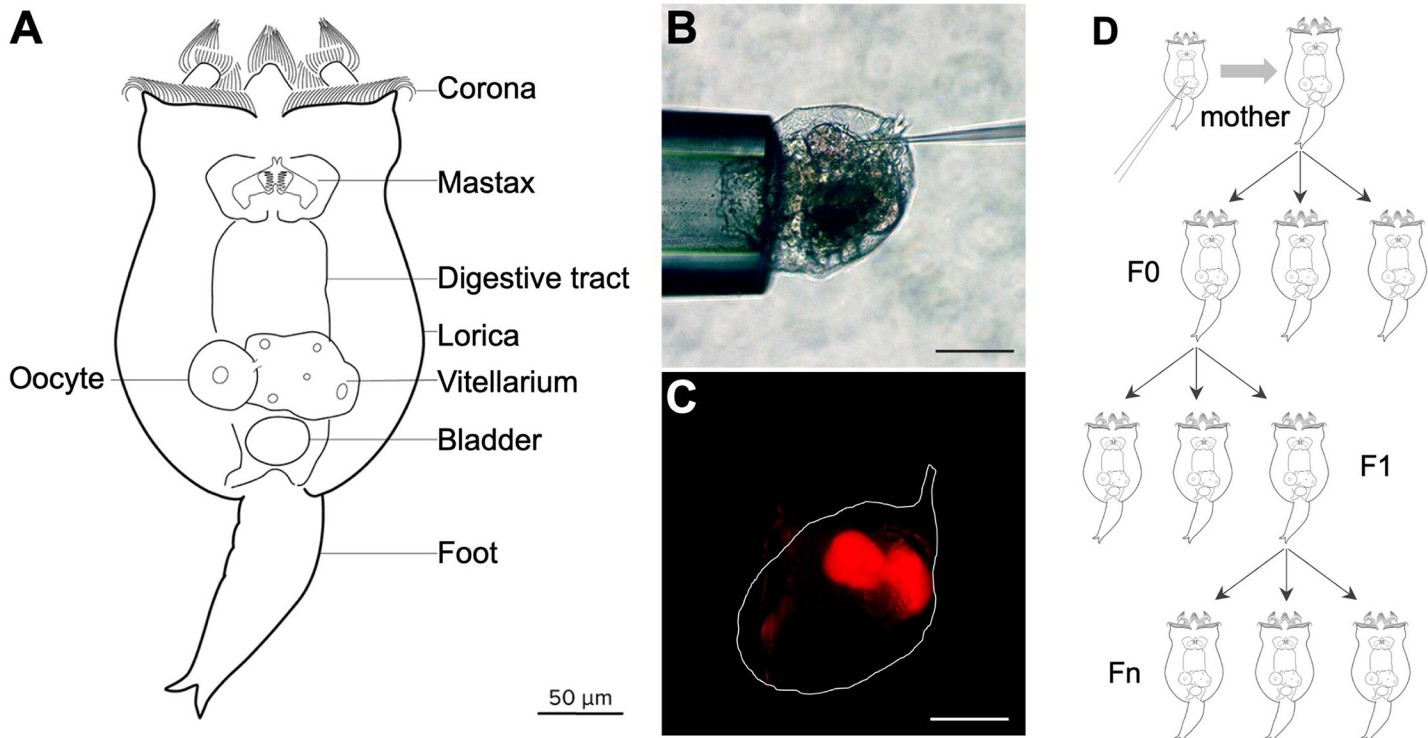

**Fig 1. Anatomy and microinjection of *Brachionus manjavacas*.** (**A**) Schematic of *B. manjavacas*. Rotifers have about 1,000 nuclei in highly syncytial tissue including muscle, digestive, nervous, and reproductive systems. The vitellarium provides material including organelles and mRNA to oocytes developing within the ovarium. Oocytes expand in size to become single-celled embryos that are extruded from the mother as eggs before further development. Multiple single-celled eggs may remain attached to the mother by thin filaments during development until just prior to hatching. Within the embryo, the vitellarium and ovarium develop from a common progenitor after the 16-cell stage and are separated from the somatic tissue by a barrier called the follicular layer [23,24]. (**B**) Microinjection: A neonate is immobilized by light suction to the apical corona while an injection needle is inserted through the integument into the vitellarium/ovarium. Scale bar, 100 μm. (**C**) Same individual as (**B**), showing fluorescence of tetramethylrhodamine in the vitellarium. Scale bar, 100 μm. (**D**) Strategy for isolating mutant lines: Injected neonate matures to an asexually reproducing female ("mother," top). Cas9 complexes are transmitted from the mother's vitellarium into oocytes that mature into the F0 generation. Many of these are expected to be mosaic [25,26]. Individuals are used for genotyping or allowed to reproduce asexually to create the F1 generation. Offspring of the F1 and subsequent generations are used for genotyping or allowed to reproduce asexually, forming clonal Fn lines; a portion of each Fn line is used for genotyping while the line is maintained through asexual reproduction.

a short region containing the CRISPR target, cloning the amplicons, and sequencing individual clones (Fig 1).

## CRISPR/Cas9-induced mutations of *vasa*

There are no known phenotypes associated with specific gene mutations in rotifers. We selected *vasa* as our initial target gene because previous studies in *Brachionus* demonstrated that *in situ* hybridization produced a distinctive expression pattern confined to the posterior of the developing ovary, allowing at least an indirect inference of function [27]. The vasa protein is a DEAD box RNA helicase with broad roles in the development and maintenance of germ cells in other animals [28,29]; thus, we predicted that knocking out *vasa* could affect offspring viability.

We selected a sgRNA target region 5′ to the DEAD box helicase motif that would cause a complete loss of function if CRISPR/Cas9 produced an indel resulting in a reading frameshift (Fig 2A). High-resolution melt (HRM) curve analysis was used to screen for mutants. Those samples for which HRM results suggested a likely mutation were genotyped by cloning and sequencing. F0 individuals showed mutations consistent with CRISPR/Cas9 activity 5′ of the

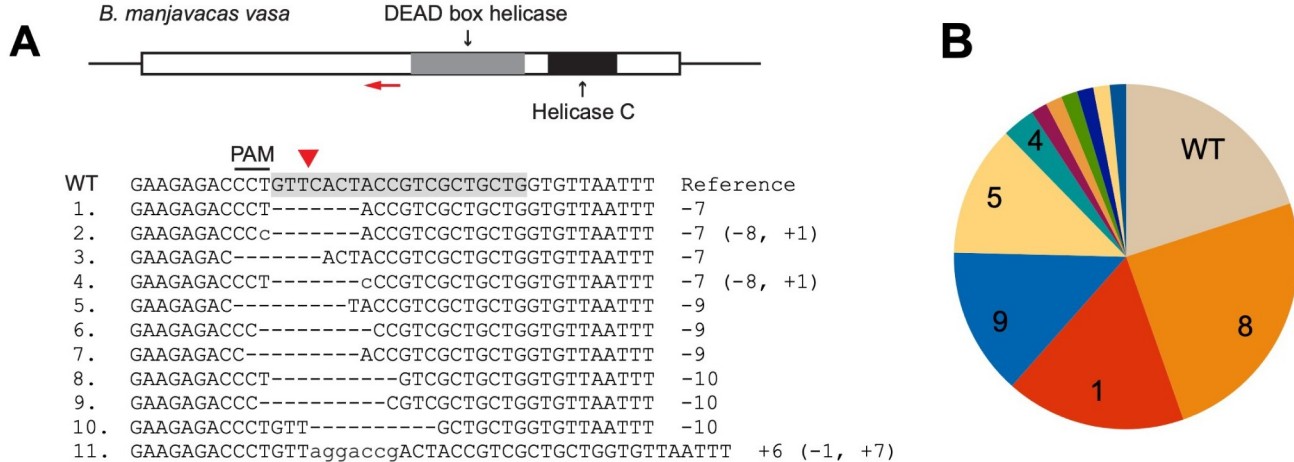

**Fig 2. CRISPR/Cas9-mediated mutagenesis of *vasa*.** (**A**) *Brachionus manjavacas vasa* gene, showing the single exon with predicted functional domains, with the sgRNA target site labelled by a red arrow. The WT reference sequence shows the sgRNA target sequence in shadow and its adjacent PAM. The Cas9-cutting site is indicated by a red triangle. Mutant alleles (from no. 1 to 11) identified in F0s and F1s of CRISPR-injected females are listed below, with deletions indicated by dashes and insertions in lowercase letters. The net indels (− deletion, + insertion, in bp) are noted on the right. (**B**) Portion of F0 and F1 individuals with each mutant allele from 4 CRISPR-injected mothers. Numbers correspond to mutation types shown in (**A**). PAM, protospacer adjacent motif; sgRNA, single-guide RNA; WT, wild-type.

targeted PAM (protospacer adjacent motif) site: Sequencing 13 clones from 1 F0 revealed a single, mutated allele, and sequencing 22 clones from a pool of 4 F0s revealed 4 additional variants (Fig 2A). No sequenced clone contained a wild-type allele, indicating a high efficiency of CRISPR activity.

Genotyping individual F1s derived from a second set of F0s showed the same mutated sequences and, at much lower frequency, a small number of additional variants, demonstrating the spectrum of mutations resulting from CRISPR/Cas9 activity (Fig 2A and 2B): 7 to 10 bp deletions starting within or immediately before the PAM site, occasionally including mismatches, and a low frequency of insertions. Medium-size deletions were facilitated by 2 bp microhomology within the target site within approximately 10 bp distance (e.g., the CT sequences at the 5′ and 3′ ends of the deletion in mutant types #1 and #4; Fig 2A). This suggests that a variety of repair mechanisms occur following Cas9 endonuclease activity, including nonhomologous end joining (NHEJ), with a bias in mechanism causing the same set of mutations to arise repeatedly in replicate experiments.

We examined the number and types of allelic variants by sequencing cloned amplicons from each of 19 F1 individuals. Each F1 had at least 1 mutation. With the caveat that we sequenced on average only 10 clones from each individual, 3 F1s appeared to be heterozygous with wild type, and 3 were heterozygous for 2 mutant alleles, with the remainder harboring more than 2 sequences (including wild type). While some of these sequences may be the result of PCR error, it appears that some F1 individuals are mosaic. The presence of up to 4 unique mutation types in individual F1s suggests editing of the germline or somatic cells in the developing F1.

We observed transgenerational CRISPR knockout of *vasa*. The population of each clonal lineage dramatically declined and became extinct with 18 days of the first F1 hatching. This suggested that *vasa* may be a haploinsufficient maternal effect gene required for development in rotifers and that the CRISPR/Cas9-induced mutations in these experiments caused lack of function sufficient to stop development and reproduction. The failure of F2s to develop and hatch demonstrates successful editing of *vasa* via CRISPR activity in the F1.

## CRISPR/Cas9-induced mutations of *mlh3*

We next set out to knock out a gene not required for normal development, but the loss of which could be deleterious under certain conditions. The MutL homolog *mlh3* meets these requirements. In model systems from yeast to mice, the Mlh3–Mlh1 protein heterodimer plays a redundant role in DNA mismatch repair process in mitosis but is critical for resolving double Holliday junctions into crossovers in meiosis [30]. Our prediction was that an *mlh3* knockout would be viable in mitotically reproducing amictic females but would interfere with meiosis in mictic females, resulting in the absence or reduced presence of males or of viable resting eggs.

We selected an sgRNA target region upstream of the predicted Mlh1-interacting protein box [31] (Fig 3A). Among 14 F0s sampled from the 3 mothers that survived injection to

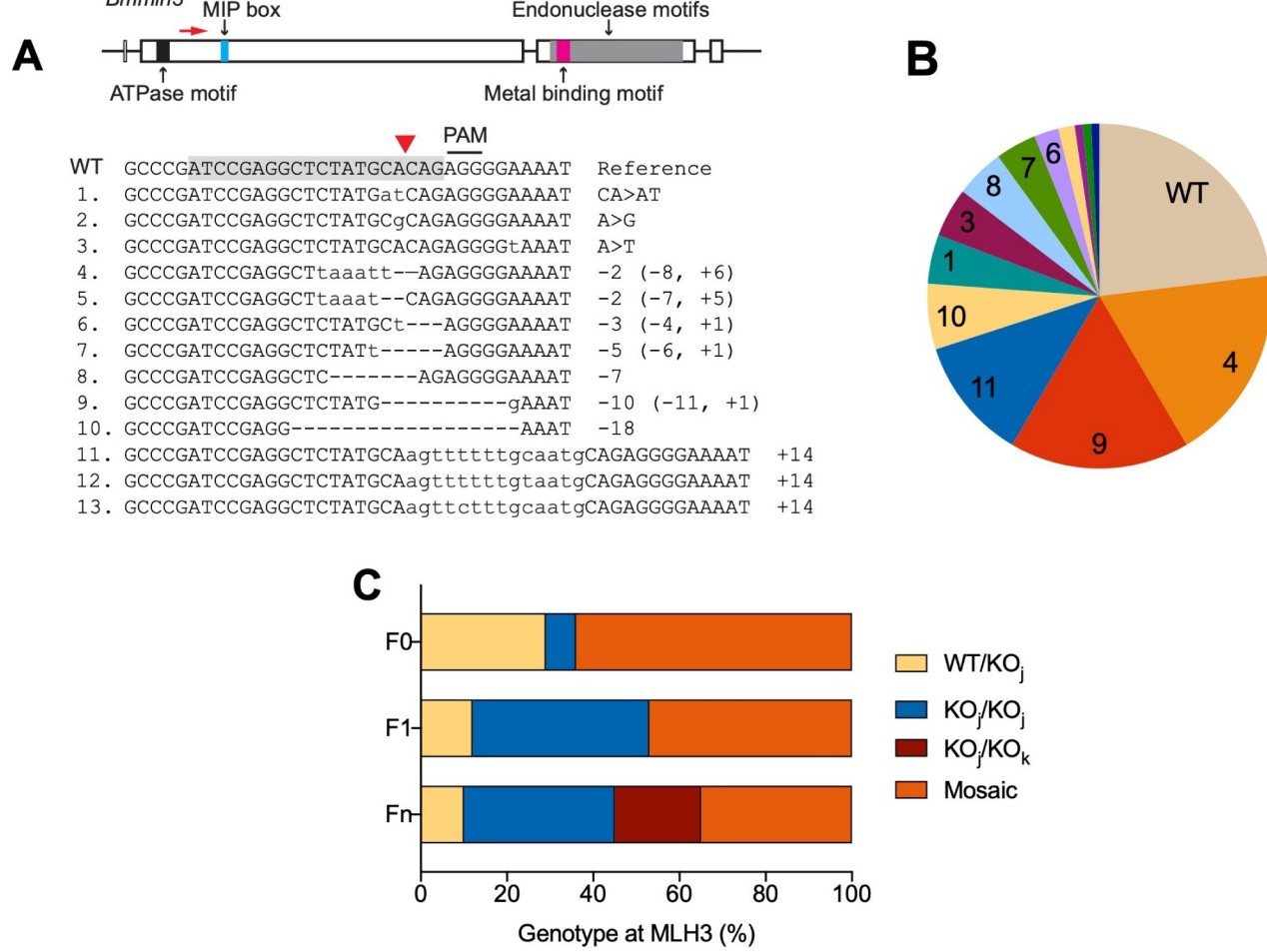

**Fig 3. CRISPR/Cas9-mediated mutagenesis of *mlh3*.** (**A**) *Brachionus manjavacas mlh3* gene, showing exons as rectangles, introns and untranslated regions as lines. Predicted functional motifs are filled with different colors. MIP box stands for Mlh1-interacting protein box. The sgRNA target site in exon 2 is labelled by red arrow. The WT reference sequence shows the sgRNA target sequence (shaded) and its adjacent PAM. The Cas9-cutting site is indicated by a red triangle. Mutant alleles (numbered 1–13) identified in the offspring of CRISPR-injected females are listed below, with deletions indicated by dashes and insertions in lowercase letters. The net indels (− deletion, + insertion, > substitution, in bp) are noted on the right. (**B**) Portion of offspring with each mutant allele from 3 CRISPR-injected mothers. Numbers correspond to mutation types shown in (**A**). (**C**) Frequency of genotypes of *mlh3* in F0, F1, and Fn of CRISPR-injected mothers: WT/KOj, heterozygous wild type and knockout; KOj/KOj, homozygous knockout; KOj/KOk, heterozygous knockout; or mosaic (3 or more alleles); see S1 Data for details of allele sequences found in each individual. The relative frequency of mosaicism decreases with each successive generation after injection. PAM, protospacer adjacent motif; sgRNA, single-guide RNA; WT, wild-type.

produce offspring, all contained at least 1 mutated *mlh3* allele, consistent with the high efficiency of CRISPR activity observed for *vasa*. One F0 appeared to be homozygous for a single CRISPR mutation, 4 appeared to be heterozygous with wild type, and the remaining 9 were mosaic. Of the 17 F1 individuals sampled from 13 F0s, 7 appeared to be homozygous for a single CRISPR mutation, and 2 appeared to be heterozygous with wild type, with the remainder appearing to be mosaic. Of the 20 Fn pools examined from 9 F0s, 7 appeared to be homozygous for a single CRISPR mutation, 2 appeared to be heterozygous with wild type, and 4 appeared to contain 2 different mutant alleles (Fig 3C). Genotyping F0 and F1 individuals revealed a broader spectrum of mutations than what we observed at the *vasa* locus, with 1 to 2 bp substitutions, medium-sized deletions of 2 to 11 bp with insertions of 0 to 6 bp, and larger insertions of 14 bp (Fig 3A and 3B). The deletion–insertion events suggest that after the Cas9-induced double strand break, the NHEJ repair process occurs at a few specific positions within 10 bp from the cut site and may involve fill-in of apparently random nucleotides. Another process may be involved to produce the larger insertions. A BLAST search with the 14-bp insertion sequence (#11 in Fig 3A) found 126 matches in the *B. manjavacas* genome, many of which were in variable number tandem repeats (VNTRs), present in at least 8 loci. CRISPR-mediated insertion comprised of repetitive sequences has been reported in mice, though the underlying repair mechanism is unknown [32]. This suggested that, in rotifers, the DNA repair process following CRISPR activity is complicated and that in addition to local sequence context, topological interactions with other regions in the genome should be considered when assessing CRISPR mutation results.

## *mlh3* knockout phenotypes

We established 6 clonal lineages homozygous for alleles CA>AT, −2 bp, −5 bp, −7 bp, −10 bp, and −18 bp. Over the course of 10 weeks and 20 to 30 generations in continuous crowded conditions that normally produce high proportions of mictic females and haploid males, we never observed males in lineages with loss-of-function alleles (−2, −5, −7, and −10 bp). Males were occasionally observed in lineages with CA>AT and −18 bp alleles. The loss-of-function mutations result in translation termination before functional motifs of Mlh3. In contrast, CA>AT substitution results in 1 amino acid change (A>D), and the −18-bp mutation causes a deletion of 6 amino acids (LYAQRG) before the MIP box. Thus, Mlh3 proteins produced by CA>AT and −18 bp mutants may still have limited function. These results suggest that functional *mlh3* is required for meiosis in sexually reproducing *B. manjavacas* and demonstrates that CRISPR-mediated knockouts can be used to study the relationship between genotype and phenotype in *Brachionus*.

While all 6 homozygous lineages were stable, 6 of the 9 mosaic F0 mutants had severely reduced fecundity, producing only 2 to 3 offspring per individual. This is 90% lower than the typical lifetime reproductive output of 25 to 30 offspring per individual in wild-type *B. manjavacas* and in the homozygous knockouts. These individuals accumulated many undeveloped eggs in their ovaries that were not extruded. As a result, the mass of undeveloped eggs deformed other structures inside the rotifer and changed the overall morphology of the posterior. Their F1 offspring (15 in total) were sterile and had a tiny vitellarium with 1 or 2 small, undeveloped eggs in the ovary. Most of these F1s died with a tiny ovary, but 2 developed a similar morphology as their mother.

## CRISPR/Cas9-mediated knock-in at the *mlh3* locus through homology-directed repair

To assess the feasibility of precisely editing the *B. manjavacas* genome through homology-directed repair (HDR), we designed a template consisting of a standard 35 nt stop codon

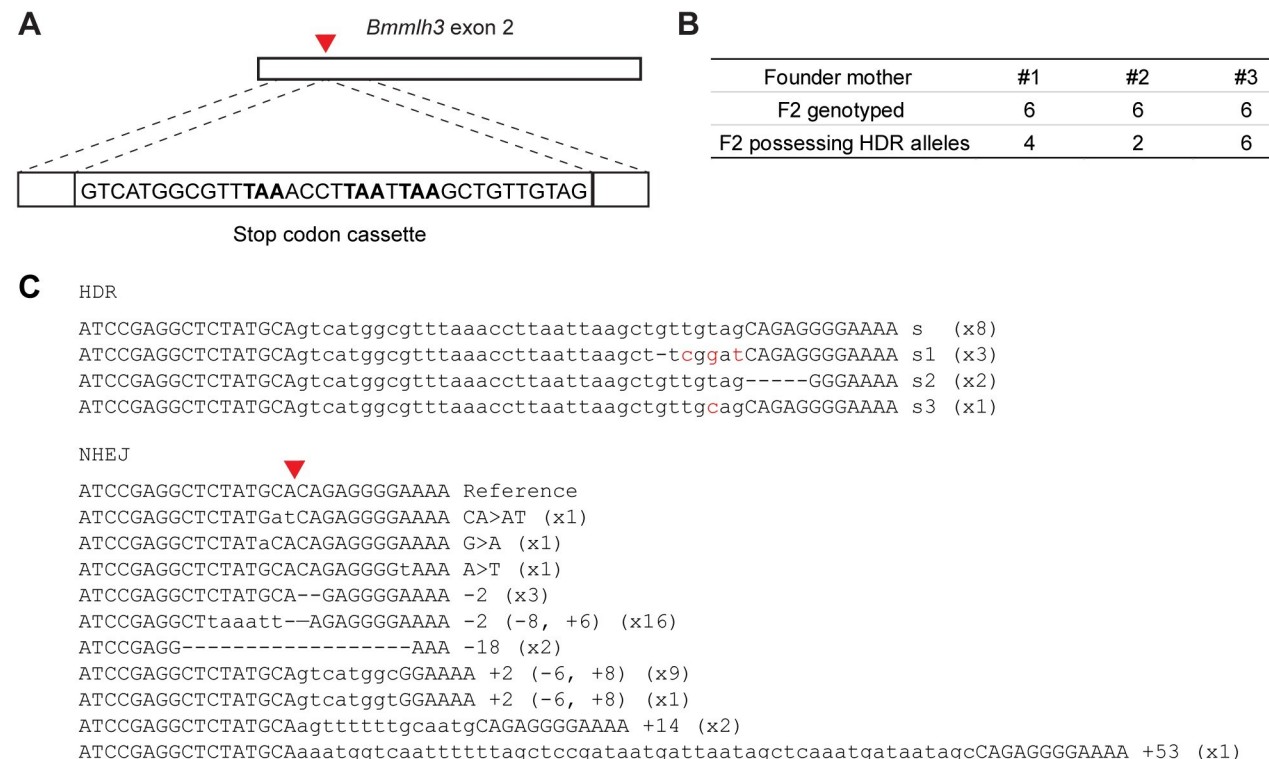

**Fig 4. CRISPR/Cas9-mediated HDR of *mlh3*.** (**A**) Schematic of ssDNA repair template containing a stop codon cassette (with stop codons in bold), flanked by 20 bp homology arms. Dashed lines indicate regions of sequence homology between the template and the *mlh3* locus. Red triangle indicates Cas9-cutting site. (**B**) The number of F2 offspring genotyped that possess HDR alleles descended from 3 injected mothers. (**C**) Mutant alleles generated by CRISPR through HDR or NHEJ. The allele with accurate stop codon cassette insertion (lowercase letters) is the top sequence, followed by 3 variants that were also found. Mutant alleles generated through NHEJ are also listed. Out of 18 F2 rotifers genotyped, the number of rotifers that possess each allele is noted on the right in parentheses. HDR, homology-directed repair; NHEJ, nonhomologous end joining; PAM, protospacer adjacent motif; ssDNA, single-stranded DNA.

cassette, which contains stop codons in all 3 reading frames, and two 20 nt homology arms that match the sequences flanking the PAM site targeted in the knock-in experiment (Fig 4A). We used the same sgRNA as in the knockout experiment and single-stranded DNA (ssDNA) complementary to the nontarget strand as the repair template. This design was based on findings from mammalian cells that Cas9 releases the PAM-distal nontarget strand first [33] and from success in zebrafish using 20 nt homology arms [34]. In an initial trial, we pooled 6 F0s from 1 CRISPR-injected mother for qPCR, cloning, and sequencing and found that approximately 8% of sequenced clones had a stop codon cassette insertion with most other clones containing indels created by NHEJ.

To determine if these HDR alleles were stably inherited, we isolated 6 CRISPR-injected mothers and their F0 and F1 offspring. Lineages from 3 mothers produced F1 males and were not genotyped based on the results described above. We genotyped 6 F2 offspring from each of the lineages from the 3 other mothers. Not surprisingly, most mutations were indels generated by NHEJ, many of which had been identified in the previous knockout experiment. However, out of 18 F2s, 12 possessed an HDR allele (Fig 4B). More than half of HDR-mediated insertions had the correct stop codon cassette; the remainder had a few erroneous nucleotides incorporated towards the 3′ end of the stop codon cassette or a small deletion in the PAM-proximal homology arm (Fig 4C). All 3 mothers produced offspring with precise and imprecise inserts of the cassette. Together, these results indicate that HDR insertions can be achieved

relatively easily in rotifers, with knock-in mutations occurring in oocytes from about half of injected mothers.

## Discussion

Here, we establish a protocol for efficient and rapid CRISPR/Cas9-mediated gene editing resulting in heritable mutation in the rotifer *B. manjavacas*. The mutations created by CRISPR/Cas9 through microinjection into the vitellaria of *B. manjavacas* can be stably inherited. Homozygous mutants account for approximately 40% of F1 individuals. The high efficiency of generating homozygous and double knockouts, asexual reproduction alleviating the need for genetic crosses, and a short asexual generation time of 3 days allow stable, transgenic mutant lineages of *B. manjavacas* to be established in about 3 weeks.

Using the method developed here, we demonstrate the use of CRISPR/Cas9-induced knockouts to examine gene function in *B. manjavacas*. Phenotypes of knockout lineages suggested that in rotifers, *vasa* may be an essential maternal effect gene and that *mlh3* may be required for meiotic production of males. However, additional, focused studies are necessary to support these preliminary observations. Stable mutant lines of *B. manjavacas mlh3* and other gene knockouts in which sexual reproduction cannot be induced will be useful to explore the mechanisms of cyclical parthenogenesis, a life history strategy common among aquatic microscopic invertebrates [21].

Finally, we demonstrate that homology-directed knock-in mutations can be generated in *B. manjavacas* with high efficiency, an application that is inefficient or lacking in many experimental systems. The length of the inserted stop codon cassette is similar to those of small epitope tags (e.g., FLAG, HA, His, and Myc) and fluorescent complementation peptides (e.g., HiBiT and GFP11). Insertion of gene reporters like these will allow examination of the timing and localization of gene expression in mechanistic studies of development, aging, and plasticity in *Brachionus* species. The efficiency of precise insertion will depend on the sequence composition and length of each insert and thus may differ from what we observe here.

The gene editing method we developed uses equipment standard in most microinjection facilities and thus can be widely adopted. Community use will lead to further optimization of the protocol, e.g., through the use of microfluidic chambers to increase the success rate of injection and the number of neonates that can be injected at very early ages, before DNA replication begins in the germline. The use of commercially synthesized gRNAs may further improve maternal survival and editing efficiency and precision. Injecting younger neonates, and titration of sgRNA and Cas9 concentrations and injection volumes, may decrease mosaicism and the spectrum of mutant types. However, given the high efficiency with which the existing protocol produces CRISPR-edited mutations and the ability to establish and phenotype clonal mutant lines through asexual reproduction, screening to eliminate mosaics is easier than screening to identify mutants in many other systems. Of course, as with any experiment using CRISPR to dissect molecular mechanisms, in rotifers or other species, screening for off-target effects will be necessary for individual experiments.

CRISPR/Cas9-mediated gene editing has been developed in only a few protostomes beyond arthropods and nematodes: an annelid worm [35], freshwater snail [36], and pygmy octopus [37]. To our knowledge, in Gnathifera, a clade of primarily microscopic aquatic species that includes rotifers, no advanced genetic manipulation tools are available for routine experimental use. Development of transgenerational gene editing for *Brachionus* thus not only expands the capabilities to conduct mechanistic research in rotifers but also broadens the capacity for comparative biology approaches across distantly related taxa. Our protocol can be adapted for use in other rotifer species that are employed for diverse studies of microevolution,

neurobiology, DNA repair, novel genetic markers, and production of antimicrobial and anti-parasitic compounds [3,38–43]. While additional work is needed to develop and assess clonal lineages for specific gene mutations in *Brachionus*, this protocol provides a template for creating novel mutants and transgenics in rotifers. This method will enable other laboratories to use rotifers to address both new and long-standing biological questions in an ecologically important and experimentally tractable animal.

## Materials and methods

### Experimental model

The monogonont rotifer *Brachionus manjavacas* Fontaneto, Giordani, Melone and Serra, 2007 (i.e., the Russian strain of the *Brachionus plicatilis* species group) was cultured in filter sterilized 15 ppt Instant Ocean (IO) artificial seawater at 21°C on a 12-h/12-h light/dark cycle and fed with the chlorophyte alga *Tetraselmis suecica*, which was maintained in bubbled f/2 medium under the same temperature and light conditions.

### sgRNA target design, ssDNA synthesis, and Cas9 protein

Sequences for *vasa* and *mlh3* were extracted from the *B. manjavacas* whole genome assembly GCA_018683815.1. Potential sgRNA targets were identified using the CHOPCHOP online server (https://chopchop.cbu.uib.no) [44] and selected to optimize GC content (40% to 60%) and the presence of microhomology flanking the Cas9-cutting site. The potential for off-target binding was examined by searching the genome of *B. manjavacas* for sequence identity to the prospective sgRNA sequences using BLAST. Candidates without other significant BLAST hits, particularly those approximately 10 bp upstream of PAM sites, were chosen as candidate sgRNA targets. The sgRNAs were synthesized following Schier's Cas9 protocol [34]. The ssDNA repair template was synthesized as a custom oligo by IDT (Table 1). EnGen Spy Cas9 NLS protein (20 μM) was purchased from NEB. The injection mixture consisted of 600 ng/μl Cas9 protein, 300 ng/μl sgRNA, (3 μM ssDNA for knock-in), 100 ng/μl tetramethylrhodamine labelled dextran, and 1× NEB buffer r3.1 and was assembled at room temperature.

### Neonate preparation and ovary injection

Eggs were collected in a 6-well plate by forcefully pipetting adult rotifers through P1000 micropipette tips to separate the eggs from mothers. Eggs were then transferred to IO containing *T. suecica* as a food source and left overnight to hatch. The next day, neonates were transferred for 3 to 5 min to Protoslo quieting solution (Carolina Biological Supply) diluted 3 times in IO or to 1 mM bupivicaine to stop their movement and then briefly washed in IO before being placed in a 60-mm petri dish filled with fresh IO. Microinjection was performed at 100× on a

**Table 1. Oligonucleotides.**

| | |
|---|---|
| *vasa* sgRNA Forward | TGTAATACGACTCACTATAGCAGCAGCGACGGTAGTGAACGTTTTAGAGCTAGAAATAGC |
| *mlh3* sgRNA Forward | TGTAATACGACTCACTATAGATCCGAGGCTCTATGCACAGGTTTTAGAGCTAGAAATAGC |
| Universal reverse scaffold | AAAAGCACCGACTCGGTGCCACTTTTTCAAGTTGATAACGGACTAGCCTTATTTTAACTTGCTATTTCTAGCTCTAAAAC |
| *mlh3* ssDNA repair template | TCCAAAAATTTTCCCCTCTGCTACAACAGCTTAATTAAGGTTTAAACGCCATGACTGCATAGAGCCTCGGATCGG |
| *vasa* HRM Forward | CCGAAAATGAACCAGTTAA |
| *vasa* HRM Reverse | CGCTGCTATTACCAGATA |
| *mlh3* HRM Forward | AAAATTGGAAAACCCTTG |
| *mlh3* HRM Reverse | AACTATTTCAACCCTGTC |

Zeiss Axio Observer using a XenoWorks Digital Microinjector (Sutter Instruments). Injection needles (1.0 mm quartz capillaries) were pulled on a P-2000 Pipette Puller (Sutter Instruments) with the parameters of heat 800, filament 4, velocity 60, delay 150, and pull 175. After pulling, needles were beveled for 15 s at a 20˚ angle using a BV-10 Micropipette Beveler (Sutter Instruments). A holding pipette made from 1.0 mm borosilicate capillary with an opening size of approximately 140 μm was used to hold the corona of the neonate. The injection needle penetrated the vitellarium through the lorica from the posterior roughly at an angle between 20˚ and 30˚. We used an injection time of 1 s to inject approximately 30 pL, which resulted in the vitellarium expanding slightly during injection. This slight expansion proved to be a practical way to judge that an injection was successful, which could then be confirmed by observing tetramethylrhodamine fluorescence. After injection, the neonate was transferred to a new 6-well plate with *T. suecica*.

### Genotyping and phenotyping the offspring of CRISPR-injected rotifers

Genomic DNA was extracted from potential mutants and genotyped by PCR, cloning, and sequencing. Individual offspring (F0s, F1s, and Fns) of CRISPR-injected rotifers were collected by snap freezing in liquid nitrogen, and genomic DNA was extracted following a modified quick embryo DNA preparation protocol. Briefly, a rotifer was incubated in 25 μl lysis buffer (10 mM Tris–HCl (pH 8.3), 50 mM KCl, 3‰ NP40, 3‰ Tween 20) at 98˚C for 10 min. Then, 2.5 μl Proteinase K (10 mg/ml) was added and incubated at 55˚C for 1 h before inactivation at 98˚C for 10 min. HRM analysis was performed to screen for mutations: each 20 μl reaction contained 2 μl genomic DNA, 0.5 μM of each primer (Table 1), 0.2 mM of each dNTP, 2.5 mM $MgCl_2$, 1× Cheetah buffer, 0.05 U/μl Cheetah Taq DNA polymerase, 1.25 μM EvaGreen Dye, and 0.625 μM ROX reference dye (Biotium). The qPCR reaction was performed using ABI StepOne Real-Time PCR System with cycle parameters of initial denaturing at 95˚C for 2 min; 50 cycles of amplification at 95˚C for 10 s, 50˚C for 15 s, 72˚C for 30 s; followed by final melting at 95˚C for 30 s and reannealing at 60˚C for 1 min. HRM disassociation curves spanned from 65˚C to 95˚C with a continuous temperature increment of 0.3%. The reannealing produced heteroduplex DNA from mosaic or heterozygous mutants, or homoduplex DNA from homozygous mutants, which has a different melt curve or a shift of melting temperature compared with WT. The PCR products that showed an obvious difference in HRM were cloned into pGEM-T easy vector (Promega) and sequenced by the Sanger method at the DNA Sequencing and Genotyping Facility at the University of Chicago Comprehensive Cancer Center.

To characterize reproductive phenotypes, F0s and their descendants were observed for up to 30 generations for changes in lifetime reproductive output, morphology of the reproductive system, and the production of mictic female and male offspring.

### Quantification of mutation types

The spectrum of mutation types of the individuals in Figs 2 and 3 were based on sequencing from 219 colonies and 371 colonies, respectively. Most of the homozygous mutants were based on 8 colonies with the same mutation types. The phenotypes of *mlh3* mutants in Fig 3 were determined from 6 individual F0s and 15 individual F1s. The allele frequencies of *mlh3* knock-in mutants in Fig 4 were based on 261 colonies.

### Supporting information

**S1 Data. Data in support of Fig 3C.** Alleles found in each individual; numbers refer to sequences in Fig 3A.
(XLSX)

## Acknowledgments

We thank the University of Chicago DNA Sequencing and Genotyping Facility and the Marine Biological Laboratory Genome Editing Core Facility for assistance.

## Author Contributions

**Conceptualization:** Kristin E. Gribble, David B. Mark Welch.

**Formal analysis:** Haiyang Feng.

**Funding acquisition:** Kristin E. Gribble, David B. Mark Welch.

**Investigation:** Haiyang Feng, Gemma Bavister, Kristin E. Gribble, David B. Mark Welch.

**Methodology:** Haiyang Feng, Gemma Bavister, Kristin E. Gribble, David B. Mark Welch.

**Project administration:** Kristin E. Gribble, David B. Mark Welch.

**Resources:** Kristin E. Gribble, David B. Mark Welch.

**Supervision:** Kristin E. Gribble, David B. Mark Welch.

**Validation:** Haiyang Feng.

**Writing – original draft:** Haiyang Feng.

**Writing – review & editing:** Haiyang Feng, Kristin E. Gribble, David B. Mark Welch.

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
