## [Editor Report · Decision Letter 0]

23 Oct 2022

Dear Dr Mark Welch, 

Thank you for submitting your manuscript entitled "Development of Highly Efficient CRISPR-Mediated Gene Editing in the Rotifer Brachionus manjavacas" for consideration as a Methods and Resources article by PLOS Biology.

Your manuscript has now been evaluated by the PLOS Biology editorial staff, as well as by an academic editor with relevant expertise, and I am writing to let you know that we would like to send your submission out for external peer review.

Once your full submission is complete, your paper will undergo a series of checks in preparation for peer review. After your manuscript has passed the checks it will be sent out for review. To provide the metadata for your submission, please Login to Editorial Manager (https://www.editorialmanager.com/pbiology) within two working days, i.e. by Oct 25 2022 11:59PM.

Kind regards,

Richard

Richard Hodge, PhD

Associate Editor, PLOS Biology

rhodge@plos.org

PLOS

---

## [Decision Letter · Decision Letter 1]

19 Nov 2022

Dear Dr Mark Welch,

Thank you for your patience while your manuscript "Development of Highly Efficient CRISPR-Mediated Gene Editing in the Rotifer Brachionus manjavacas" was peer-reviewed at PLOS Biology. I'm handling your paper temporarily while my colleague Dr Richard Hodge is out of the office. It has now been evaluated by the PLOS Biology editors, an Academic Editor with relevant expertise, and by three independent reviewers. 

In light of the reviews, which you will find at the end of this email, we would like to invite you to revise the work to thoroughly address the reviewers' reports.

IMPORTANT: You'll see that while the reviewers are tentatively positive about your study, they each raise a number of concerns that must be addressed before further consideration. I discussed these requests with the Academic Editor, who thought that the concerns were reasonable, and pointed out that you may be able to "kill two birds with one stone" by using editing of an additional gene to address the principal concerns of both reviewers #2 and #3.

Specifically, the Academic Editor said: "I think the comments and suggestions from the reviewers are very good. The suggestion with F3 is doable, since they have a very fast generation time. Maybe give them a bit longer for the major revision. And indeed, they will need to show it with another gene, since vasa effects the germ line. It will become very convincing if they do this. Also the repeat with another gene will satisfy reviewer #3, since they can also exclude one more time the non-specific effects. The gene should be chosen wisely, but the authors know what to do best. They have the knowledge."

Based on the Academic Editor's advice, I'm giving you an additional month (4 months instead of our "COVID-normal" 3 months) to perform the work, but do get back to us if you need more.

Given the extent of revision needed, we cannot make a decision about publication until we have seen the revised manuscript and your response to the reviewers' comments. Your revised manuscript is likely to be sent for further evaluation by all or a subset of the reviewers.

We expect to receive your revised manuscript within 4 months. Please email us (plosbiology@plos.org) if you have any questions or concerns, or would like to request an extension. 

**IMPORTANT - SUBMITTING YOUR REVISION**

*Re-submission Checklist*

*Published Peer Review*

*PLOS Data Policy*

*Blot and Gel Data Policy*

Sincerely,

Roli Roberts

Roland G Roberts PhD

Senior Editor

PLOS Biology

rroberts@plos.org

on behalf of

Richard Hodge, 

Associate Editor

PLOS Biology

rhodge@plos.org

REVIEWERS' COMMENTS:

Reviewer #1:

Title: Development of Highly Efficient CRISPR-Mediated Gene Editing in the Rotifer Brachionus

manjavacas

Authors: Haiyang Feng et al.

Authors made CRISPR/Cas9-mediated knock out mutants with an application for knock-in technology by homology-directed repair using rotifers. Their success rate was 50-66% of the injected neonates survived to produce offspring. This is not bad if we consider rotifers are doing parthenogenesis.

However, the interesting researchers need to approach the genome information to determine off-target binding sites for Brachionus manjavacas. However, they are not published its genome into journal. This is a main flaw of this manuscript.

The Discussion section is too short and they need to discuss more substantially to the better understanding for the readers.

Also in Fig. 2, the authors need to show HMA and % of mutant spectra of deleted mutants (should be reached to almost 100%) but they did not show it in the manuscript. 

Overall, I cannot support to publish this manuscript to PLoS Biology.

Reviewer #2:

The study be Feng et al., is a promising and exciting advance for CRISPR/Cas9 genome editing in the rotifer Brachionus manjavacas, an important animal system for examining molecular mechanisms of metazoan evolution. It will be of broad, significant interest to investigators using non-model system aquatic species for evolution and ecotoxicology, and more generally as another example of the universality of CRISPR/Cas9 genome editing across biological systems. The authors show robust targeted mutagenesis at two genes, vasa and mlh3, by injection of CRISPR/Cas9 RNPs in to the vitellaria of adult females. CRIPSR/Cas9 targeted knock-in of stop codon sequences from a single stranded oligonucleotide template was demonstrated at mlh3. The data indicates the edited alleles are transmitted through F1 animals to the F2 generation. Although CRISPR/Cas9 gene editing has previously been reported in somatic tissues of the rotifer Brachious koreanus using electroporation (Kim et al., 2019), the current study by Feng et al., has very high novelty in demonstrating effective germline CRISPR/Cas9 gene editing and knock-in via injection into the vitellaria/oocytes of Brachionus manjavacas.

Overall the study was well organized but seems somewhat preliminary, and the presentation of how transmission of edited alleles is followed through the germline to the F1 and F2 generations could have been more clearly explained. As presented the data does not demonstrate the establishment of stable, clonal mutant or precision knock-in lines, as claimed in the introduction. In genetic model systems, a family of F3 animals is generated from a single F2 individual transmitting an edited allele, to establish a mutant line carrying the stable germline allele. It was unclear if this is a feasible goal in this rotifer, since gene edited animals were inviable and infertile in the F1 and F2 generation. The paper would be strengthened by establishing an F3 rotifer line that has a specific edit in the genome, and the line used for robust genetic analysis linking phenotype to genotype. If this is the longer term intent it could be better explained in the Discussion. 

Other issues:

1. The very strong claims presented in the Discussion should be tempered to indicate further studies are needed to show indel and precision knock-in tagged alleles can be generated and used for mechanistic studies, and generation of reporter knock-ins is a possibility.

2. Other suggested improvements to strengthen the paper include optimizing targeted mutagenesis by titration of CRISPR/Cas9 reagents. Only one condition for CRISPR/Cas9 mutagenesis was tested, and only a percentage of the targeted animals survived. There were resulting phenotypes including fertility and viability that were attributed to the targeted genes, but these phenotypes as indicated may be due to off targeting or gene knock-out by biallelic inactivation. Targeted knock-in of a 3Xstop cassette using an ssoligo template led to the identification of precise and imprecise alleles in F2 animals. Additional detail showing the lineage of alleles from each F1 would clarify the frequency of recovering precise integrations - not every F0 animal may transmit a precise knock-in allele. Having an idea of the number of F0 adults that need to be screened to recover a precise allele in the F1 to F2 generation is useful for replicating this approach. 

3. Figure 1. The authors begin the results section with a clear description of the approach used for CRISPR/Cas9 gRNA RNP targeting of oocytes in the adult female vitellaria, and Figure 1 illustrates the approach. This is very much appreciated. 

It would also be very helpful if there were a diagram of a flow chart of allele analysis through the F0 - F1 - F2 generations. A panel in Figure 1 showing life cycle or expected allele transmission (asexual or sexual) would be useful. A single F0 could transmit multiple alleles through the germline if the germline cells continue to be edited during early F0 development - this is established in other animal systems. 

Inclusion of the amount/molarity of reagents injected, either in the diagram, text, or figure legend would be helpful, particularly since this a novel approach for CRISPR/Cas9 targeting in rotifers. The methods state the concentration of reagents in the injection mix, but not the amount injected.

For future analyses, more consistent targeted mutagenesis may be achieved with synthetic guide RNAs from IDT or Synthego, which provide quality control, reducing possible toxicity due to the variable integrity of in-house synthesized and purified gRNAs.

4. Figure 2 shows the results of indel targeting to vasa to generate lof alleles. In Figure 2B plot it was unclear what the y axis and x axis represent. Is the x axis (1 - 11) the individual 11 mutant alleles shown in 2A? and the y axis is the number of times that allele appeared in individual F1s? 

In Figure 2C, it seems there are 19 individual F1s represented on the x-axis. The y-axis is labeled mutagenesis ratio. The arrow pointing to Control - is this meant to be a value of zero? Many of the individual F1s noted on the x-axis show multiple alleles. 

Pg. 7 lines 154-156: The text states "ongoing CRISPR activity carried from F0s to F1 oocytes or embryos" is responsible or F1s with 2+ alleles, and incorrectly cites references 24 (animal zygote targeting) and 25 (editing in arabidopsis). This is highly unlikely - CRISPR/Cas9 RNP particles would not be stable throughout the development of an F0 animal and segregated into its germline and F1 embryos. The citation # 24 by Mehravar et al., describes mosaicism in the F0 zygote targeted generation. Please remove this line and these references. F1 animals from a single F0 may inherit different germline alleles, but an individual F1 animal can only inherit, at most, two germline alleles. 

The data indicated a stable germline vasa mutant couldn't be established since F1s died within 18 days. The text states this demonstrates a maternal effect, but to definitively show this, mothers that are homozygous for a mutant allele, generated by fertilization of haploid gametes carrying an established allele, is necessary. 

5. Figure 3 shows the results of targeting the non-essential mismatch repair mutL Homolog mlh3. 

Panel 3D shows a mosaic F1 with overall reduced morphology and few oocytes, but the genotype of this individual isn't indicated. Again it isn't clear how an individual F1 could carry more than two alleles. 

It could be helpful to target another gene not involved in DNA repair - disruption of mlh3 activity could have potential negative impact on genome replication during embryonic growth/cell divisions, leading to reduced viability and fertility. Targeting a developmental transcription factor involved in lineage specification, so that a stable line could be isolated, would be interesting.

6. Figure 4. F2 animals were recovered carrying precise and imprecise knock-in alleles of STOP codons, from an ss oligo template, at the mlh3 target site. From 6 injected mothers, 3 had F1 offspring transmitting, and F2 inheriting, knock-in alleles. A total of 18 F2s were genotyped - a record of which of the 3 mothers transmitted a precise allele to F1 and F2 would be useful to know. This would clarify how many F0 animals need to be raised to recover a precise knock-in allele transmitted to the next generation.

Reviewer #3:

This paper introduces an important new method that opens up functional genetics in the phylum Rotifera for the first time. Rotifers are used in lots of areas of research in part because they are small animals with fast generation times. I agree with the authors that current work is hampered by the lack of ways to alter genes to test their functional effects. Knowledge of the deep branching clade of animals that rotifers belong to is limited at present because of the lack of tractable genetic model systems. Two knockouts and one knock-in are presented. 

One challenge due to the lack of previous tools is that good genetic/phenotypic markers (like eye colour in Drosophila) are not known already. Two genes with different presumed effects are chosen, and the primary method for gauging success is cloning and sequencing. It might have been interesting to see how expression level of vasa varies among the different mutant types to confirm that different amounts of the vasa gene are expressed depending on the pattern of mutation introduced. Some phenotypic observations are presented that fit with expectations, although both with a complication that wasn't predicted a priori (e.g. there is partial viability of vasa knockouts, irrespective of whether wt is present or not,; for mlh3 there is an apparent effect during amixis as well as during mixis). Is there any further evidence that can be used to confirm this is an unexpected effect of gene knockout as opposed to other unanticipated effects of the the CRISPR manipulation (which might be continuing activity between generations)? 

In an ideal world, you might prefer clearer cut links from genotype to phenotype for the initial test of the method, but I think this encapsulates the severe challenges of the state of play in rotifer research. This manuscript paves the way for future studies that introduce gene reporters and systematically investigate other genetic pathways. 

Some more comments are listed below.

Line 75. It would be good to add half a sentence saying why affecting the germ line is key for the benefits outlined above. 

Line 93. It might be clear to everyone, but worth highlighting these are all amictic still here?

Fig 1C. I can't see much here except a bi-lobed red blob. Would it be permissible to highlight the outline of the animal to aid interpretation?

Line 112-115. I don't follow why it is useful to have a gene expressed in the vitellarium and oocytes, as your method is targeting DNA rather than RNA? Somehow it seemed to imply that it is good for the gene to be expressed in the place that you are injecting…? Also, you say it is expected to affect offspring viability - this seems an odd choice if you're wanting to detect transmission of the edit in the F1. 

Line 159. If you think there was ongoing CRISPR activity, could it be that the decline was caused by other mutational events rather than haploidy of the vasa gene. For example, could your injected Cas9 recruit other RNA in the cells and cause different mutations?

Line 190. Intriguing. Were any of these repeats close to your target region or are they far apart in the genome?

Line 235. Interesting additional effect of the manipulation. Is it 100% certain that 'amictic' egg production in Brachionus is really mitotic rather than automictic? (I imagine it is but mention just in case). Or again, could this be off-target residual effects of the Cas9?

Line 261. Regarding the knock-ins, did you see any phenotypic effects of these?

Line 281 "Allows stable transgeneration lineages in about 3 weeks" - I thought they all became extinct after 18 days? Which cases survived for 3 weeks?

A diagram of the design for each knockout and the knock in with number of individuals in F0 and F1 might be useful to have an overview of the design. I found it quite hard to keep track

---

## [Decision Letter · Decision Letter 2]

31 May 2023

Dear Dr Mark Welch,

Thank you for your patience while we considered your revised manuscript "A Method for Highly Efficient CRISPR-Mediated Gene Editing in the Rotifer Brachionus manjavacas" for publication as a Methods and Resources Article at PLOS Biology. This revised version of your manuscript has been evaluated by the PLOS Biology editors, the Academic Editor and two of the original reviewers.

Based on the reviews, I am pleased to say that we are likely to accept this manuscript for publication, provided you address the following data and other policy-related requests that I have provided below (A-C):

(A) We would like to suggest the following modification to the title: 

“Highly efficient CRISPR-mediated gene editing in a rotifer”

(B) You may be aware of the PLOS Data Policy, which requires that all data be made available without restriction: http://journals.plos.org/plosbiology/s/data-availability. For more information, please also see this editorial: http://dx.doi.org/10.1371/journal.pbio.1001797

-Supplementary files (e.g., excel). Please ensure that all data files are uploaded as 'Supporting Information' and are invariably referred to (in the manuscript, figure legends, and the Description field when uploading your files) using the following format verbatim: S1 Data, S2 Data, etc. Multiple panels of a single or even several figures can be included as multiple sheets in one excel file that is saved using exactly the following convention: S1_Data.xlsx (using an underscore).

-Deposition in a publicly available repository. Please also provide the accession code or a reviewer link so that we may view your data before publication. 

Figure 1C

(C) Please also ensure that each of the relevant figure legends in your manuscript include information on *WHERE THE UNDERLYING DATA CAN BE FOUND*, and ensure your supplemental data file/s has a legend.

We expect to receive your revised manuscript within two weeks. 

*Published Peer Review History*

*Press*

Kind regards,

Richard

Richard Hodge, PhD

Associate Editor, PLOS Biology

rhodge@plos.org

Reviewer remarks:

Reviewer #2: The authors have addressed all issues.

This reviewer appreciates the author's explanation that an F0 could transmit more than two alleles to offspring. It makes sense, given the rapid development and short generation time of rotifers, and persistence of Cas9 RNP, Casa9 RNP could continue to mutagenize cells in the vitellaria and germline stem cells ovarium.

---

## [Editor Report · Decision Letter 3]

9 Jun 2023

Dear David,

On behalf of my colleagues and the Academic Editor, Andreas Hejnol, I am pleased to say that we can accept your manuscript for publication, provided you address any remaining formatting and reporting issues. These will be detailed in an email you should receive within 2-3 business days from our colleagues in the journal operations team; no action is required from you until then. Please note that we will not be able to formally accept your manuscript and schedule it for publication until you have completed any requested changes.

PRESS

Best wishes, 

Richard

Richard Hodge, PhD

rhodge@plos.org

PLOS
